# Successive POI Recommendation via Brain-inspired Spatiotemporal Aware Representation

## Abstract

Existing approaches usually perform spatiotemporal representation in the spatial and temporal dimensions, respectively, which isolates the spatial and temporal natures of the target and leads to sub-optimal embeddings. Neuroscience research has shown that the mammalian brain entorhinal-hippocampal system provides efficient graph representations for general knowledge. Moreover, entorhinal grid cells present concise spatial representations, while hippocampal place cells represent perception conjunctions effectively. Thus, the entorhinal-hippocampal system provides a novel angle for spatiotemporal representation, which inspires us to propose the SpatioTemporal aware Embedding framework (STE) and apply it to POIs (STEP). STEP considers two types of POI-specific representations: sequential representation and spatiotemporal conjunctive representation, learned using sparse unlabeled data based on the proposed graph-building policies. Notably, STEP jointly represents the spatiotemporal natures of POIs using both observations and contextual information from integrated spatiotemporal dimensions by constructing a spatiotemporal context graph. Furthermore, we introduce a user privacy secure successive POI recommendation method using STEP, and it achieves the state-of-the-art performance on two benchmarks. In addition, we demonstrate the excellent performance of the STE representation approach in other spatiotemporal representation-centered tasks through a case study of traffic flow prediction problem. Therefore, this work provides a novel solution to spatiotemporal aware representation and paves a new way for spatiotemporal modeling-related tasks.

## 1 Introduction

With the rapid growth of location-based web services like Instagram and Yelp, there has been a seismic shift in how people interact with locations around them. Through exploitation of Points-of-Interest (POIs) and their contexts, successive POI recommendation can benefit users and businesses greatly. As a core of POI information utilization, encoding POIs into vector representation space is of great significance for advanced POI analysis and downstream applications. Existing studies attempt to represent POI from different perspectives and collaborate with user preference modeling to achieve recommendation. Since consecutive check-ins are usually highly correlated, naturally, sequence modeling approaches like the Markov chain model were used to capture the check-in sequential characteristics of POIs (Ye et al., 2011; Liu et al., 2013; Zhang, 2014; Feng et al., 2015). Employing tensor factorization technique, the works (Yang et al., 2017; Wang et al., 2018) modeled target users and POIs separately by interacted features for POI recommendation. More recently, enlightened by neural networks' success, recurrent neural nets were remolded to represent POIs and user preferences implicitly (Liu et al., 2016a; Zhao et al., 2019; Zhu et al., 2017). Considering the geographical attributes of POIs, researchers have used power-law distribution, Gaussian distribution, or hierarchical tiling methods to depict the geographical influence over POI distributional features (Ye et al., 2011; Lian et al., 2014; Feng et al., 2017; Chang & Kim, 2020; Luo et al., 2020). However, geographical modeling methods above only provide single-scale or coarse-grained manually designed representations of POI geographical influences, which is deficient in capturing the POI-specific spatial features. Also, the arbitrary modeling might even lead to over-parameterization. While temporal dimension offers indeterminate auxiliary information for POI modeling, to utilize

the POI temporal information within the check-ins, some works using time interval, time state variables or temporal transition vectors to promote the POI representing (Zhao et al., 2019; 2016; 2017; Li et al., 2018; Manotumruksa et al., 2018; Zhao et al., 2020). However, these methods focused on utilizing general temporal patterns among all POIs and failed to exploit the POI-specific visiting time patterns sufficiently. Still, the POI-specific spatiotemporal characteristics were not adequately mined and utilized.

**Inspirations.** The entorhinal-hippocampal system plays a central role in the mammal cognition architecture. The Nobel Prize-winning neuroscience research (O'keefe & Nadel, 1978) demonstrated that entorhinal grid cells provide an effective multi-scale periodic spatial representation (Yuan et al., 2015; Banino et al., 2018; Mai et al., 2020; Dang et al., 2021). Moreover, the entorhinal-hippocampal system is also critical for the non-spatial inference that relies on understanding the associations between perceptions from various perspectives (Whittington et al., 2018; Stachenfeld et al., 2018; Whittington et al., 2020). Some promising researches cast spatial and non-spatial problems as connected graphs and point out the cells inside entorhinal-hippocampal structure provide efficient conjunctive representation for those graphs (Stachenfeld et al., 2018; Gustafson & Daw, 2011). As the representation mechanism in the entorhinal-hippocampal system was extensively studied, it is widely accepted that conjunctions of representations from different aspects form the hippocampal representation for relational memory (Whittington et al., 2018; 2020; Eichenbaum, 2017; MacDonald et al., 2011; Sargolini et al., 2006). For the general spatiotemporal aware embedding, various contexts can be constructed into affinity graphs for latent representation learning. Furthermore, strategies like conjunctive-representing in entorhinal-hippocampal structure can be translated to improve the quality of the representations (see Fig.1 left part).

In this paper, borrowing inspirations from the entorhinal-hippocampal system, we propose the SpatioTemporal aware Embedding framework, namely STE and apply to POIs (STEP) for successive POI recommendation, the model architectures are shown in Figure 1. Firstly, we build context graphs to enable unsupervised embedding learning on sparse check-ins. Secondly, we employ a sequential model to represent POIs from the check-in sequence perspective. Most importantly, we elaborate a spatiotemporal model consists of a grid-cell spatial encoder and a visiting time encoder to capture the POI-specific spatiotemporal characteristics. The spatiotemporal model learns to get the POI spatiotemporal latent representations using the spatiotemporal context graph. Finally, we implement successive POI recommendation systems based on the STEP and achieve high-performance using simple recurrent neural networks as recommenders.

This work's main contributions are summarized as follows:

(1) Motivated by the graph-representing strategy of structural knowledge in the entorhinal-hippocampal system, we solve the spatiotemporal aware embedding learning in a graph-based unsupervised learning manner through specific context graph building policies, especially the spatiotemporal context graph, to fully exploit rich unlabeled data.

(2) Inspired by the conjunctive representation mechanism in the entorhinal-hippocampal complex, we present a spatiotemporal model with a grid-cell spatial encoder and a time pattern encoder to utilize the spatiotemporal information. The conjunctive representing approach based on a unique spatiotemporal context graph addresses the problem of previous spatiotemporal modeling methods in which spatial and temporal information are isolated and represented separately.

(3) We introduce a successive POI recommendation system by incorporating STEP and simple sequence predictors to show the feasibility of implementing specific applications based on the proposed STE framework. We perform experiments on large real-world datasets to demonstrate the effectiveness of STEP, and our method outperforms baselines according to experimental results.

(4) Moreover, compared with classical recommendation systems, our POI-centered solution can avoid the ethical risks of artificial intelligence, like personal data leakage, as it does not need access to private information such as user preferences. Furthermore, our framework can be applied to more valuable applications like wildlife preservation and urban traffic scheduling as a general spatiotemporal aware modeling method.

## 2 PRELIMINARIES

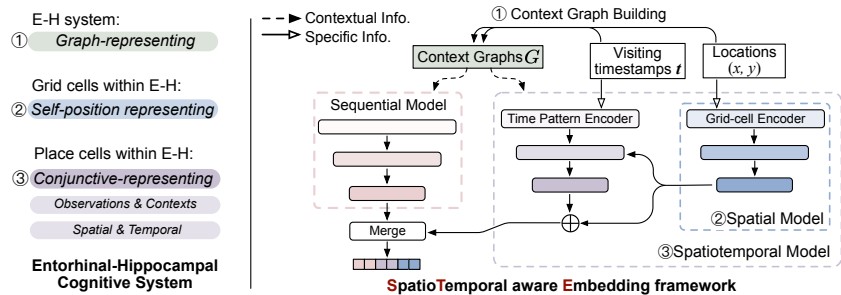

Figure 1: Representing mechanisms in entorhinal-hippocampal system (*E-H system* for short) and the framework of spatiotemporal aware embedding model. The proposed STE framework consists of three essential parts: the context graph building strategies to construct simplified affinity graphs, the spatiotemporal model to extract rich item-specific spatiotemporal features, and the sequential model to extract sequential feature embeddings. The uniqueness about our spatiotemporal information usage is that we represent item from spatiotemporal perspective (not isolated) using observations and contexts conjunctively.

Given a set of POIs with corresponding coordinates $\mathcal{P} = \{p_i\}, p_i = (x_i, y_i)$, a check-in sequence is one set of continuous check-ins of one user in one day, denoted as $\mathcal{S}_j = \{(p_1, t_1^1), ..., (p_n, t_m^n)\}$. Unlike previous works, we do not regard all check-in records of a user as one sequence since check-ins with relatively long intervals are not very relevant. Although we assign notation to users for generality, the user information is not used in the training phase except to split sequences.

Table 1: Notations in this paper.

| Notation | Definition |
|---|---|
| $t_j^i$ | $j$-th timestamp of $p_i$ |
| $\boldsymbol{t}^i$ | Visiting time pattern matrix of $p_i$ |
| $\boldsymbol{e}_{spa}^i$ | Spatial vector representation of $p_i$ |
| $\boldsymbol{e}_{seq}^i$ | Sequential vector representation of $p_i$ |
| $\boldsymbol{e}_{st}^i$ | Spatiotemporal conjunctive representation of $p_i$ |
| $\boldsymbol{e}_{step}^i$ | STEP vector representation of $p_i$ |
| $\oplus$ | Tensor concatenation operation |

We define context graphs as graphs that encode context information as affinity among POIs. Various contexts in the check-in records can be easily built into graphs $G_p = \{V_p, E_p\}$, where $V_p$ is the set of POIs and $E_p$ is the set of edges between adjacent POIs. The edges in context graphs represent the correlation between neighboring POIs defined by geographical distance, relative position in check-in sequences, or spatiotemporal adjacent criterions. We summarize notations in this paper using Table 1.

**Data description.** The Instagram Check-in dataset (Chang et al., 2018) was collected from Instagram in New York and the data was preprocessed in the same manner as previous works (Zhao et al., 2016; 2017). The Instagram Check-in dataset has been pre-processed when it is made public, it includes 2,216,631 check-in records at 13,187 POIs of 78,233 users. Check-in sequences are sorted by timestamps, the first 70% are used as a training set and the remaining 30% for validation and testing. The Gowalla dataset is a globally-collected large-scale social media dataset (Cho et al., 2011). We eliminate users with fewer than ten check-ins and POIs accessed by fewer than ten users. Then the check-in records are sorted according to timestamps and first 70% check-ins are used for training and the remaining latest records for testing. We perform vivid data analyses in the Appendix due to the space constraints.

## 3  SPATIOTEMPORAL AWARE EMBEDDING MODEL OF POIS

We illustrate components of STEP: the sequential model in Sec. 3.1, the spatiotemporal model in Sec. 3.2, and state the STEP-based successive POI recommendation method in Sec. 3.3. We adopt a *simple-minded (no-parameters) edge weighting policy* for constructing all context graphs. Weight $A_{i,j} = 1$ if vertices $i$ and $j$ is connected, this simplification avoids the necessity of choosing edge-weighting parameters.

### 3.1 SEQUENTIAL MODEL

The sequential sub-model represents POIs using context graph $G_{seq}$. Given one POI and its context in the check-in sequence, entry $A_{i,j}$ in adjacency matrix of $G_{seq}$ is 1 if $p_i, p_j$ are within the same context window. This is a common way to mine the sequential correlations of tokens like words (Mikolov et al., 2013) or POIs (Lim et al., 2020). Our sequential model aims to predict true contextual POIs, *i.e.*, connected vertices in $G_{seq}$. Intuitively, minimizing the objective function over all target-neighbor pairs guarantees that POIs sharing similar sequential context will have shorter distances in embedding space (Hadsell et al., 2006). To avoid the intractable summation over the whole context space, we follow the noise contrastive sampling approach (Gutmann & Hyvärinen, 2012; Mikolov et al., 2013) to get an approximated surrogate loss

$$\mathcal{L}_{seq}(\theta_{seq}) = - \sum_{p_i, p_j \in \mathcal{P}} \left[ \mathbb{I}(\gamma = 1) \log \sigma(\boldsymbol{e}_{seq}^i \cdot \boldsymbol{e}_{seq}^j) + \mathbb{I}(\gamma = -1) \log \sigma(-\boldsymbol{e}_{seq}^i \cdot \boldsymbol{e}_{seq}^j) \right], \quad (1)$$

where $\gamma = 1$ if $(p_i, p_j)$ is a sequential neighboring pair and $\gamma = -1$ if not, indicator $\mathbb{I}$ outputs 1 when the argument condition is true and otherwise 0. This unsupervised loss can also be seen as taking expectation with respect to the distribution $\mathbb{P}(p_i, p_j, \gamma)$ over $\mathcal{P}$, which is conditioned on the POI sequential context graph $G_{seq}$.

### 3.2 SPATIOTEMPORAL MODEL

In this section, we illustrate the POI spatiotemporal conjunctive embedding model in detail. The proposed spatiotemporal model is composed of two key components: a POI spatial model and a POI visiting time encoder, an intuitive illustration can be found in Fig. 1.

#### 3.2.1 SPATIAL MODEL

The spatial sub-model takes location observations $(x_i, y_i)$ and spatial context graph $G_{spa}$ to produce spatial representations.

**Grid-cell encoding.** Inspired by the multi-scale periodic representation of grid cells in mammals, we formulate our POI spatial contextual encoder to use sinusoidal and cosinusoidal functions of different scales to encode the raw locations of POIs in geographical space following previous works Gao et al. (2019); Mai et al. (2020). Given a POI $p_i = (x_i, y_i) \in \mathbb{R}^2$, the grid-cell model based encoder encodes the coordinates in 2-D Euclidean space into spatial latent representations in $\mathbb{R}^{d_{spa}}$. We denote the grid cell encoder based spatial embedding of POI $p_i$ as

$$\boldsymbol{e}_{spa}^i = \phi(\psi(x_i, y_i); \theta_{spa}), \quad (2)$$

where

$$\psi(x_i, y_i) = \psi^1(x_i, y_i) \cdots \oplus \psi^s(x_i, y_i) \cdots \oplus \psi^S(x_i, y_i) \quad (3)$$

is concatenated multi-scale representations of $6S$ dimensions, $S$ denotes the number of grid scales and $\phi$ represents fully connected non-linear layers. Considering three unit vectors $\boldsymbol{a}_1=[1,0]^{\mathrm{T}}$, $\boldsymbol{a}_2=[-1/2, \sqrt{3}/2]^{\mathrm{T}}$, $\boldsymbol{a}_3=[-1/2, -\sqrt{3}/2]^{\mathrm{T}} \in \mathbb{R}^2$, at each frequency, position codes

$$\psi^s(x_i, y_i) = \psi_1^s \oplus \psi_2^s \oplus \psi_3^s \quad (4)$$

are computed via

$$\psi_k^s(x_i, y_i) = \left[ \cos(\frac{[x_i, y_i] \cdot \boldsymbol{a}_k}{\rho \lambda_{\min}}), \sin(\frac{[x_i, y_i] \cdot \boldsymbol{a}_k}{\rho \lambda_{\min}}) \right], k \in \{1, 2, 3\} \quad (5)$$

and $\rho = (\lambda_{\max}/\lambda_{\min})^{s/(S-1)}$. $\lambda_{\min}$ and $\lambda_{\max}$ are the minimum and maximum scale values, here we use $S = 64$ following the previous work (Mai et al., 2020) and set $\lambda_{\max} = 1km$, $\lambda_{\min} = 0.1km$.

**Spatial-neighboring definition.** We project the coordinates in the geographical coordinate system WGS84 to the projection coordinate system NAD27 to get locations of POIs in $\mathbb{R}^2$. For each entry $A_{i,j}$ in adjacency matrix of spatial context graph $G_{spa}$, we assign $A_{i,j}$ using the geographical distances. Specifically, we computed the geographical distances between POIs and construct an undirected spatial context graph $G_{spa}$ with *uniform edges* among the *top-ten closest* POIs (nearest neighbors policy). As the grid cell encoder can handle geographical distributions at different scales

(Mai et al., 2020), we do not use a specific radius ($\epsilon$-neighborhoods policy) to filter the neighboring POIs to fully exploit the multi-scale representation capability. The spatial graph construction process is related to Lim et al. (2020), in which the edges are weighted according to average distances to enable graph attention computations.

Given a target POI $p_i$, neighboring contextual POI set $\mathcal{P}_{spa}^+$ and negative set $\mathcal{P}_{spa}^-$ sampled from $G_{spa}$, the unsupervised embedding learning can simply be maximizing the log-likelihood of observing the true context POIs. We can formulate this target with negative sampling via a general objective function:

$$\mathcal{O}(ctx) = -\sum_{p_i \in \mathcal{P}} \sum_{p_j \in \mathcal{P}_{ctx}^+} \left[ \log \sigma(\boldsymbol{e}_{ctx}^j \cdot \boldsymbol{e}_{ctx}^i) + \frac{1}{K} \sum_{p_k \in \mathcal{P}_{ctx}^-} \log \sigma(-\boldsymbol{e}_{ctx}^k \cdot \boldsymbol{e}_{ctx}^i) \right], \quad (6)$$

where $ctx$ indicates the context graph type and $ctx \in \{seq, spa, st\}$ in this work, $\sigma$ is the sigmoid function and $K$ denotes the number of samples in negative sample set $\mathcal{P}_{ctx}^-$. Following Eq.6, the loss function for the spatial context embedding model is:

$$\mathcal{L}_{spa}(\theta_{spa}) = \mathcal{O}(spa). \quad (7)$$

### 3.2.2 SPATIOTEMPORAL CONTEXT GRAPH CONSTRUCTION

For constructing spatiotemporal context graph $G_{st}$, we want to mine the item-specific spatiotemporal conjunctions, so for each entry $A_{i,j}$ of the adjacency matrix of $G_{st}$, we assign $A_{i,j} = 1$ following the hierarchy of *neighboring timestamps* → *temporal neighboring* → *spatiotemporal neghboring*:

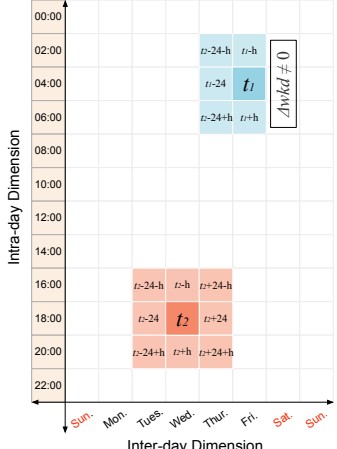

1. **Neighboring timestamps.** Given an arbitrary timestamp pair $(t_1, t_2)$, time interval $\Delta t \triangleq |t_1 - t_2|$ and $\Delta wkd \triangleq |wkd(t_1) - wkd(t_2)|$ where $wkd(t) = 1$ if $t$ is weekend else 0. For one time $t$, its temporal-neighboring timestamps are those within the neighborhood window and satisfy $\Delta wkd = 0$, as shown in Fig. 2. h is a hyper-parameter indicates temporal context window width and $h \in (0, 24)$ hours.

2. **POI temporal neighboring.** POI $p_i$ and $p_j$ with corresponding visiting timestamp sets $\mathcal{T}_i = \{t_1^i, t_2^i, \dots\}$ and $\mathcal{T}_j = \{t_1^j, t_2^j, \dots\}$. The number of neighboring timestamp pairs $(t^i, t^j)$ > m, where $t^i \in \mathcal{T}_i, t^j \in \mathcal{T}_j$, m is a threshold.

Figure 2: Temporal neighboring examples with h = 2. For $t_1$, some times are excluded from the temporal neighborhood window as $\Delta wkd \neq 0$.

3. **POI spatiotemporal neighboring.** If $p_i$, $p_j$ are spatial and temporal neighboring, they are spatiotemporal neighboring.

It is redundant for individual POI-specific temporal modeling since solely relying on time information, we are not able to recommend reasonable candidate POIs (visits may take place contemporarily all over the world). Thus, time is regarded as a supplementary dimension of basic spatial information, our spatiotemporal context graph provides an effective way to combine the POI-specific temporal and geographical information.

### 3.2.3 VISITING TIME PATTERN ENCODING

We develop a visiting time pattern encoding method to tensorise the temporal observations (visiting records in timestamps $t$) to provide observation inputs for spatiotemporal model. Unlike previous works (Liu et al., 2016a; Zhao et al., 2019; Zhu et al., 2017; Zhao et al., 2016), we focus on the POI-specific temporal patterns rather than general temporal characteristics among all timestamps. Compared with previously used time interval-based or hard-coded methods, our encoding scheme can tensorise the item-specific temporal information more precisely and able to provide a reliable decision basis for spatiotemporal modeling.

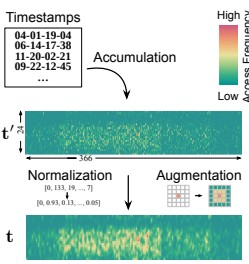

Figure 3: Schematic of the visiting time encoding process.

**Visiting time encoding.** Since the visiting time patterns are relatively

stable on the scale of year but fickle on the hour and the date scale, in our encoding scheme, visits are counted by the check-in timestamps to raw matrices $\boldsymbol{t'} \in \mathbb{R}^{24 \times 366}$. Then, the raw matrices $\boldsymbol{t'}$ are normalized and applied Gaussian kernel for smoothing, this process also reasonably augment the POI accessing time data. The visiting time encoding process is shown schematically in Fig. 3. The final matrix representation $\boldsymbol{t} = \texttt{smooth}(\texttt{norm}(\boldsymbol{t'}))$ retains the fine-grained check-in time patterns as well as rough item-specific visiting time features.

### 3.2.4 SPATIOTEMPORAL CONJUNCTIVE EMBEDDING LEARNING

We formulate the spatiotemporal conjunctive representation as:

$$\boldsymbol{e}_{st}^{i} = \phi_{\theta_{st}}(\phi_{\theta_{time}}(\boldsymbol{t}^{i}), \boldsymbol{e}_{spa}^{i}) \oplus \boldsymbol{e}_{spa}^{i}, \tag{8}$$

where $\phi$ indicates fully-connected layers. Follow the formulation in Eq.6, we implement the spatiotemporal conjunctive representation learning by minimizing:

$$\mathcal{L}(\theta_{time}, \theta_{st}) = \mathcal{O}(st). \tag{9}$$

We sample $\mathcal{P}_{st}^{+}$ and $\mathcal{P}_{st}^{-}$ from $G_{st}$, where $\mathcal{P}_{st}^{+}$ is the set of spatiotemporal-neighboring POIs whereas $\mathcal{P}_{st}^{-}$ is the set of negative POIs.

During the optimization procedure, the spatial model is jointly optimized as a sub-model of spatiotemporal model, the full objective of the spatiotemporal model is

$$\mathcal{L}_{st} = \mathcal{L}(\theta_{time}, \theta_{st}) + \lambda_{spa} \mathcal{L}_{spa}(\theta_{spa}), \tag{10}$$

$\lambda_{spa}$ is a weighting factor for preserving the spatial context information during the spatiotemporal modeling. We first sample a batch of spatial context $G_{spa}$ to optimize the spatial context loss $\mathcal{L}_{spa}$ to preserve geographical context. Next we sample a batch of spatiotemporal context $G_{st}$ to optimize the spatiotemporal loss $\mathcal{L}_{st}$ to preserve the spatiotemporal context. We repeat above procedures for $I_0$ and $I_1 = I_0 / \lambda_{spa}$ iterations respectively to approximate the balancing factor $\lambda_{spa}$. We update all parameters $\{\theta_{spa}, \theta_{time}, \theta_{st}\}$ of spatiotemporal model in iterations until the overall loss $\mathcal{L}_{st}$ converges.

### 3.3 SUCCESSIVE POI RECOMMENDATION WITH STEP

We present the STEP (STE of POIs)-based recommendation method in detail in this section. Taking the spatiotemporal data as input, we construct context graphs $G$ and feed the observations (locations and time patterns) into the STEP model to perform embedding learning. The embeddings are smoothed according to corresponding context graphs to preserve contextual information. Then, POI vector representations (embeddings) $\boldsymbol{e}_{seq}, \boldsymbol{e}_{st}$ are merged as spatiotemporal aware embedding $\boldsymbol{e}_{step}$ and fed into the recommender to generate an estimated embedding $\hat{\boldsymbol{e}}$. Specifically, we use concatenation

$$\boldsymbol{e}_{step}^{i} = \boldsymbol{e}_{seq}^{i} \oplus \boldsymbol{e}_{st}^{i} \tag{11}$$

to merge the two sequential and spatiotemporal embeddings. This merging policy can preserve information from different spaces without extra-parameters and not requires the embeddings to be in the same dimension (*e.g.* $d_{st} = d_{seq}$) thus provides more flexibility. We adopt two-layer recurrent networks as the recommender model.

**Embedding model optimizing.** Parameters in the STEP embedding model are optimized according to corresponding loss function $\mathcal{L}(\theta_*) = \mathcal{L}_* + \alpha ||\theta_*||_2$, where $\mathcal{L}_* \in \{\mathcal{L}_{seq}, \mathcal{L}_{st}\}$, $\theta_* \in \{\theta_{seq}, \Theta_{st}\}$, $\Theta_{st} = \{\theta_{spa}, \theta_{time}, \theta_{st}\}$. $\alpha$ is the weighting factor of the 2-norm regularizer.

**Recommender model optimizing.** The predictor is then optimized with the pre-trained STEP embedding model for the next POI recommendation task. During the training phase, given a $n$-length check-in sequence $\mathcal{S}_j$, we can get corresponding STEP embedding series of POIs $\{\boldsymbol{e}_{step}^{(1)}, \dots, \boldsymbol{e}_{step}^{(gt)}\}$, the last POI is regarded as the recommendation target. The target of the recommender is to predict latent representation $\hat{\boldsymbol{e}}_{step}$ similar to the embedding of true successive POI $\boldsymbol{e}_{step}^{(gt)}$, formally described as:

$$\underset{\theta_{pred}}{\arg \max} \sum_{\mathcal{S}_j \in \mathcal{S}} \texttt{sim}\left(\hat{\boldsymbol{e}}_{step}, \boldsymbol{e}_{step}^{(gt)}\right). \tag{12}$$

Table 2: Comparisons with baselines on two datasets, we mark **best** values with bold fonts and underline the suboptimal ones. CAPE-based methods are not applicable on Gowalla (no tweets were provided) and we do not report results of some methods on Instagram Check-in as they cannot be reproduced faithfully. †: with user preference consideration, ‡: using semantic content information.

| DATASET | Instagram Check-in | | | | Gowalla | | | |
|---|---|---|---|---|---|---|---|---|
| METHOD\METRIC | HIT@1 | HIT@5 | HIT@10 | MRR | HIT@1 | HIT@5 | HIT@10 | MRR |
| Random+GRU | 0.1197 | 0.2207 | 0.2726 | 0.1792 | 0.0715 | 0.0725 | 0.0732 | 0.0727 |
| Random+LSTM | 0.1207 | 0.2225 | 0.2751 | 0.1805 | 0.0722 | 0.0736 | 0.0749 | 0.0737 |
| Skip-Gram+GRU | 0.1356 | 0.2419 | 0.3040 | 0.1919 | 0.1090 | 0.2111 | 0.2617 | 0.1612 |
| Skip-Gram+LSTM | 0.1318 | 0.2344 | 0.2984 | 0.1875 | 0.1085 | 0.2101 | 0.2585 | 0.1594 |
| CAPE+GRU‡ | 0.1390 | 0.2433 | 0.3079 | 0.1953 | $N/A$ | $N/A$ | $N/A$ | $N/A$ |
| CAPE+LSTM‡ | 0.1381 | 0.2412 | 0.3054 | 0.1939 | $N/A$ | $N/A$ | $N/A$ | $N/A$ |
| Geo+GRU | 0.1619 | 0.2616 | 0.3248 | 0.2093 | 0.1267 | 0.2309 | 0.2834 | 0.1684 |
| Geo+LSTM | 0.1622 | 0.2594 | 0.3128 | 0.1875 | 0.1233 | 0.2296 | 0.2811 | 0.1701 |
| ST-RNN† | 0.1054 | 0.2019 | 0.2426 | 0.1681 | 0.0519 | 0.0953 | 0.1304 | 0.2187 |
| STGN† | − | − | − | − | 0.0256 | 0.0784 | 0.1144 | 0.0590 |
| STGCN† | − | − | − | − | 0.0424 | 0.1134 | 0.1625 | 0.0842 |
| LSTPM† | 0.1261 | 0.2134 | 0.3121 | 0.1957 | 0.1468 | 0.2506 | 0.2983 | 0.1998 |
| STP-DGAT† | − | − | − | − | 0.1344 | 0.2414 | 0.2653 | 0.1856 |
| STP-UDGAT† | − | − | − | − | 0.1475 | 0.2911 | 0.3285 | 0.2130 |
| STEP+RNN | 0.2458 | 0.3170 | 0.3502 | 0.2822 | 0.1495 | 0.2878 | 0.3634 | 0.2222 |
| STEP+GRU | **0.2467** | 0.3057 | 0.3336 | 0.2781 | 0.1490 | 0.2912 | 0.3636 | 0.2233 |
| STEP+LSTM | 0.2454 | **0.3204** | **0.3556** | **0.2835** | **0.1539** | **0.2968** | **0.3728** | **0.2282** |

The objective function of recommender is:

$$\mathcal{L}_{pred}(\theta_{pred}) = -\sum_{\mathcal{S}_j \in \mathcal{S}} \left[ \log \sigma' \left( \mathtt{sim}(e_{step}^{(gt)}, \hat{e}_{step}) \right) - \log(\sum_{p_i \in \mathcal{P}} \sigma'(\mathtt{sim}(e_{step}^i, \hat{e}_{step}))) \right], \quad (13)$$

where $\sigma' = \exp(\mathtt{LeakyReLU}(\cdot))$ and $\mathtt{sim}(\cdot, \cdot) = \frac{\mathbf{a} \cdot \mathbf{b}}{||\mathbf{a}|| \cdot ||\mathbf{b}||}$. During the testing phase, we compute the cosine similarity scores to rank the candidate POIs to generate recommendation lists.

## 4 EXPERIMENTS

We perform the successive POI recommendation task on the Instagram Check-in dataset, Gowalla dataset, and the traffic flow forecasting task on TaxiBJ15 and TaxiBJ.

Table 3: Context graphs statistics on two POI datasets.

| AVERAGE VALUE | Ins. | Gow. |
|---|---|---|
| Records per POI | 168.1 | 34.2 |
| $E_{seq}$ per POI | 66.2 | 35.5 |
| $E_{spa}$ per POI | 10.0 | 10.0 |
| $E_{st}$ per POI | 0.997 | 0.596 |

### 4.1 SUCCESSIVE POI RECOMMENDATION TASK

**Metrics.** During the system inferencing phase, the recommendation system recommends a POI list according to the estimated scores of candidate POIs for every trial sequence. We apply widely-used metrics HIT@K (if the ground truth is within the top-k of the list, a score of 1 is awarded, else 0), $k = 1, 5, 10$ and MRR (Mean Reciprocal Rank) for evaluation. These metrics reflect different aspects of the recommendation lists, HIT@K measures the rate of valid recommendation among all trials, whereas MRR scores the quality of the entire recommendation list.

**Hyper-parameter settings.** We set the hyper-parameters of our proposed method to the following default values. We set context window size in the POI sequential model to 2 and adopt $\mathtt{h} = 2$, $\mathtt{m} = 11$ for building $G_{st}$. We utilize Adam optimizer with batch size $512, \beta_1 = 0.9, \beta_2 = 0.999$ and set the initial learning rate to $0.001$ followed by a reduce-on-plateau decay policy, the decay factor is 0.1 during the training. Weighting factors $\alpha, \lambda_{spa}$ are set to $1 \times 10^{-4}, 0.2$ and the embedding dimensions $\{d_{seq}, d_{spa}, d_{st}\}$ are set to $\{32, 64, 96\}$.

We compare the STEP-based successive POI recommendation method with representative methods: **Embedding-recommender methods.** We choose six methods consisting of three embedding models and two recommenders. For the embedding model, we use the following embedding models. (1) *Random*, (2) *Skip-Gram* (Liu et al., 2016b), (3) *CAPE* (Chang et al., 2018) and (4) *Geo* (Mai

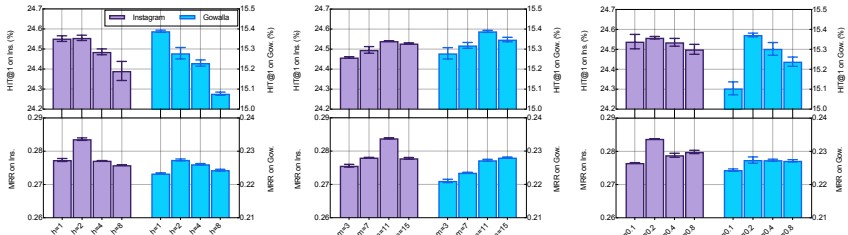

Figure 4: Effect of hyper-parameters h, m, $\lambda_{spa}$ on Instagram Check-in and Gowalla datasets. The means and standard deviations are computed over five runs using different random seeds.

et al., 2020). For the recommender model, two-layer networks based on (1) *GRU* unit (Merri & Fellow, 2014) and (2) *LSTM* unit (Hochreiter & Schmidhuber, 1997) are used. **One-stage methods.** We choose representative one-stage methods as baselines. (1) *ST-RNN* (Liu et al., 2016a). (2) *STGN* (Zhao et al., 2019), a LSTM variant which models visit preferences with time and distance considerations, and the improved variant *STGCN*. (3) *LSTPM* (Sun et al., 2020) is a LSTM-based method. (4) *STP-DGAT* and *STP-UDGAT* (Lim et al., 2020) are spatial-temporal-preference user dimensional graph attention networks.

**Comparison results.** According to results in Table 5, our method outperforms the baselines by significant margins on both datasets, and the gains in recommendation accuracy are especially substantial on the Instagram dataset with rich temporal information (according to Table 3, POIs in Gowalla have less visiting timestamps). The one-stage recurrent network-based methods, such as *ST-RNN*, *LSTPM* surpassing basic embedding-based methods by more sufficient exploitation of user-preference spatiotemporal properties. However, these methods remain noncompetitive with STEP-based ones, although the STEP stands without user preference consideration. The advantageous performance of our method over the competitors can be attributed to its efficient use of the item-specific spatiotemporal nature. We observe significant performance improvements of the STEP-based methods in terms of MRR, indicating the STEP-based methods provide better candidates lists on both datasets, benefit from the efficiency of the proposed spatiotemporal aware embedding model. The usage of LSTM units in recommender slightly improves the performance compared with basic RNN cells because of their advantages in gate functions of recurrent connections. Moreover, the basic RNN recurrent net equipped with STEP embedding can also outperform one-stage SOTAs. This also proves the effectiveness of our brain-inspired spatiotemporal aware embedding model.

**Effect of hyper-parameters.** We study the effect of newly-introduced hyper-parameters h, m, $\lambda_{spa}$, and report the results using HIT@1 and MRR in Fig.4. h and m control the sparsity of the spatiotemporal context graph and $\lambda_{spa}$ regulates the importance of the spatial context smoothness term in the spatiotemporal model objective function. We alter h to build spatiotemporal context graphs with decreasing sparsity as larger h corresponds to coarse temporal-neighboring condition. The increasing h within a certain range results in performance improvements but appears detrimental to the precise top-1 recommendation. The best performance (determined by MRR) is obtained when $h = 2$. The choice of h is task-related, according to our results on two dataset, $h = 2$ can be a good initial value for POI recommendation according to our results. This value can be further adjusted for different application scenarios or datasets to build spatiotemporal context graphs with desired sparsity. We set m from 3 to 15, larger m leads to a sparser spatiotemporal context graph. We observe the performance slightly changes after increasing the threshold m, when $m = 11$, the best performance is obtained. We also investigate the effect of the balancing factor $\lambda_{spa}$ for spatiotemporal model training, the recommendation system achieves the best performance when $\lambda_{spa} = 0.2$ and further increases only bring minor improvements, thus we select 0.2 as a default value in this work, this also helps reduce unnecessary iterations during the model training.

**Ablation study.** We study the effectiveness of STEP modules by performing successive POI recommendation task with LSTM recommender, the method using standard STEP embedding model is referred as FULL.
After the removal of (A) POI-specific time information processing module, the spatiotemporal model degenerates to a spatial model. According to Table 4, the method performs worse without (A) on both datasets. According to Table 3, POIs in Gowalla have sparser specific observation $t$s and contextual information from $G_{st}$. This results in more significant performance

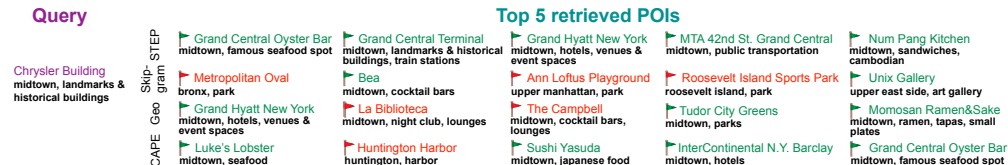

Figure 5: Retrieval results on Instagram dataset Top 5 candidates are placed from left to right. Green and red flag marks represent positive and negative POIs determined by both *opening hours* and *geographical position*. Under names of those POIs, we list their region and representative tags collected from websites with bold fonts.

improvements on the Instagram set than on Gowalla. We replace POI visiting time pattern matrices $t$ (applying Sec.3.2.3) with random-initialized matrices in $\mathbb{R}^{24 \times 366}$. We note that the use of (B) POI visiting time encoder improves the recommendation performance on both datasets, and the improvement are positively correlated with the temporal information abundance of the dataset. We use a one-layer neural network location encoder $\psi'(x, y)$ to replace (C) grid-cell encoder in the STEP to demonstrate its effectiveness. Results in Table 4 demonstrate that the grid-cell encoder improves the quality of STEP representation and leads to better successive POI recommendation performances on both datasets. We observe noticeable performance drops after the removal of (D) spatial embedding preserving in Table 4 since the spatiotemporal conjunctive representation integrate spatial and temporal attributes at the cost of spatial information loss (due to the dimensional reduction). The use of (D) spatial embedding information preserving alleviate this problem to a certain extent.

Table 4: Effectiveness of using (A) temporal information, (B) visiting time encoder, (C) grid-cell encoder, (D) spatial embedding preserving and (E) the spatiotemporal model. We mark the **best** ones with bold fonts and underline the suboptimal ones.

| | | HIT@1 | HIT@5 | HIT@10 | MRR |
|---|---|---|---|---|---|
| Instagram | FULL | 0.2454 | **0.3204** | **0.3556** | **0.2835** |
| | w/o (A) | 0.2433 | 0.2544 | 0.2607 | 0.2504 |
| | w/o (B) | 0.2452 | 0.3151 | 0.3442 | 0.2798 |
| | w/o (C) | 0.2399 | 0.3037 | 0.3305 | 0.2727 |
| | w/o (D) | 0.2298 | 0.2531 | 0.2674 | 0.2451 |
| | only (E) | **0.2466** | 0.2840 | 0.3021 | 0.2664 |
| Gowalla | FULL | **0.1539** | **0.2968** | **0.3728** | **0.2282** |
| | w/o (A) | 0.1461 | 0.2825 | 0.3540 | 0.2174 |
| | w/o (B) | 0.1509 | 0.2921 | 0.3664 | 0.2248 |
| | w/o (C) | 0.1006 | 0.2169 | 0.2922 | 0.1657 |
| | w/o (D) | 0.0973 | 0.2117 | 0.2866 | 0.1610 |
| | only (E) | 0.1252 | 0.2142 | 0.2519 | 0.1683 |

A simple LSTM recommender can achieve competitive recommendation performance *without* check-in sequential information consideration (Table 4 only(E)), demonstrating the effectiveness of the spatiotemporal model in STEP. As visiting sequential information provides relatively coarse-grained POI depictions, the removal of the sequential model even leads to HIT@1 improvement on the Instagram dataset. Also, the metric fallen on Gowalla (-18.6%, -27.8%, -32.4%, -26.2%) is more significant than those on Instagram set (+3.1%, -11.4%, -15.0%, -6.0%), exactly opposed to their temporal information abundance, quantitatively evaluated by average timestamps per POI.

## 4.2 PERFORMING TRAFFIC FLOW FORECASTING TASK WITH STE

To demonstrate the generalizability of STE, we perform traffic flow forecasting task with the proposed spatiotemporal embedding methods. The relevant content is presented in detail in Appendix.A.

## 5 CONCLUSION AND DISCUSSION

In this paper, we propose the spatiotemporal aware embedding framework STE and apply it to POIs (STEP). To the best of our knowledge, this is the first work that translating entorhinal-hippocampal representing mechanisms to the spatiotemporal embedding. Inspired by the graph-representing policy in brain entorhinal-hippocampal system, STEP captures sequential and spatiotemporal representation from unlabeled sparse data through context graph building and graph-based embedding learning. Moreover, STEP provides a highly efficient spatiotemporal model motivated by grid cells' multi-scale spatial representation and place cells' conjunctive representation, which overcomes the problem caused by frequently used separate-representing. STEP-based successive POI recommendation method outperforms baselines and SOTAs on two real datasets without user preference invasion. Furthermore, this work presents a practical framework for effective spatiotemporal aware modeling of general items, enabling more valuable spatiotemporal-related tasks such as wildlife preservation and urban traffic management.

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

## A    CASE STUDY: TRAFFIC FLOW FORECASTING WITH STE

As an essential basis of traffic scheduling, accurate traffic flow forecasting is of great significance. In this part, we perform traffic flow forecasting with the proposed STE, which is abbreviated as STE-TG (STE of Traffic Grid).

### A.1    PROBLEM FORMULATION

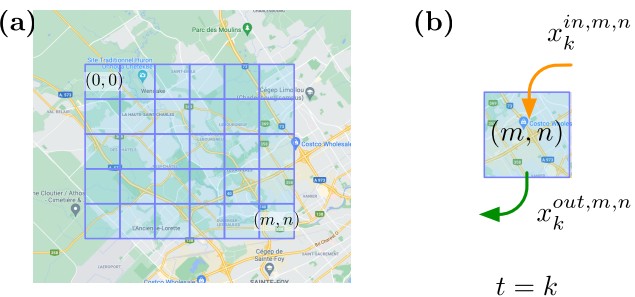

Figure 6: An illustration of (a) grid map partition and (b) flow of a grid.

Following previous works (Hoang et al., 2016; Zhang et al., 2016), we partition a city into an $M \times N$ *traffic grid* map based on the longitude and latitude where a traffic grid represents a geographical region. For a grid $(m, n)$ that lies at the $m^{th}$ row and the $n^{th}$ column, two types of traffic flows at $k^{th}$ timestamp are considered, namely inflow and outflow, defined as

$$x_k^{in,m,n} = \sum_{Tr_k \in \mathbb{P}} |\{i > 1 \mid g_{i-1} \notin (m, n) \wedge g_i \in (m, n)\}| \tag{14}$$

$$x_k^{out,m,n} = \sum_{Tr_k \in \mathbb{P}} |\{i \geq 1 \mid g_i \in (m, n) \wedge g_{i+1} \notin (m, n)\}| \tag{15}$$

where $Tr_k$ means the trajectory at the $k^{th}$ timestamps, $g_i$ indicates the geographical coordinate, $g_i \in (m, n)$ means point $g_i$ lies within grid $(m, n)$ and $|\cdot|$ means the cardinality of a set. At timestamp $k$, the traffic flow of grid $(m, n)$ is represented as $X_k^{m,n} = (x_k^{in,m,n}, x_k^{out,m,n})$. The goad of traffic flow forecasting is: given $X_k, k \in \{0, 1, ..., t - 1\}$, predict $X_t$.

### A.2    METHODOLOGY

As illustrated in Fig.1 in the main text, the input of STE-TG is composed of observations (coordinates and visiting time patterns) and contexts (context graphs). Unlike STEP, as there is no sequential traffic information, the sequential model is not applicable.

**STE usage**    Following Sec.3.2.1 in the main body of the article, we use spatial observation and build a spatial context graph in the same way. Specifically, same hyper-parameter settings of $S, \lambda_{min/max}$ is adopted in grid-cell encoder, and top-10 policy is used for building $G_{spa}$. Also, the embedding objective of the spatial sub-model is still the term in Eq.7. Different from POIs, the timestamps of the traffic grid flow data are collected using a standard time interval. Thus both the encoding of visiting time patterns and the construction of spatiotemporal context graph could be simplified. Notably, we generate the visiting time pattern matrix as the temporal observation input for the spatiotemporal model in the same way we did for POIs, and since the timestamp is standard, we skipped the accumulation step in Fig.3. For constructing $G_{st}$, we define the grid temporal adjacency using a specified distance function $d(\cdot, \cdot)$ which is calculated by the normalized traffic flow difference value of two traffic grids.

$$d((m, n), (m', n')) \triangleq \frac{1}{t} \sum_{k=1}^{t} \left[ \mathsf{abs} \left( x_k^{in,m,n} - x_k^{in,m',n'} \right) + \mathsf{abs} \left( x_k^{out,m,n} - x_k^{out,m',n'} \right) \right]. \tag{16}$$

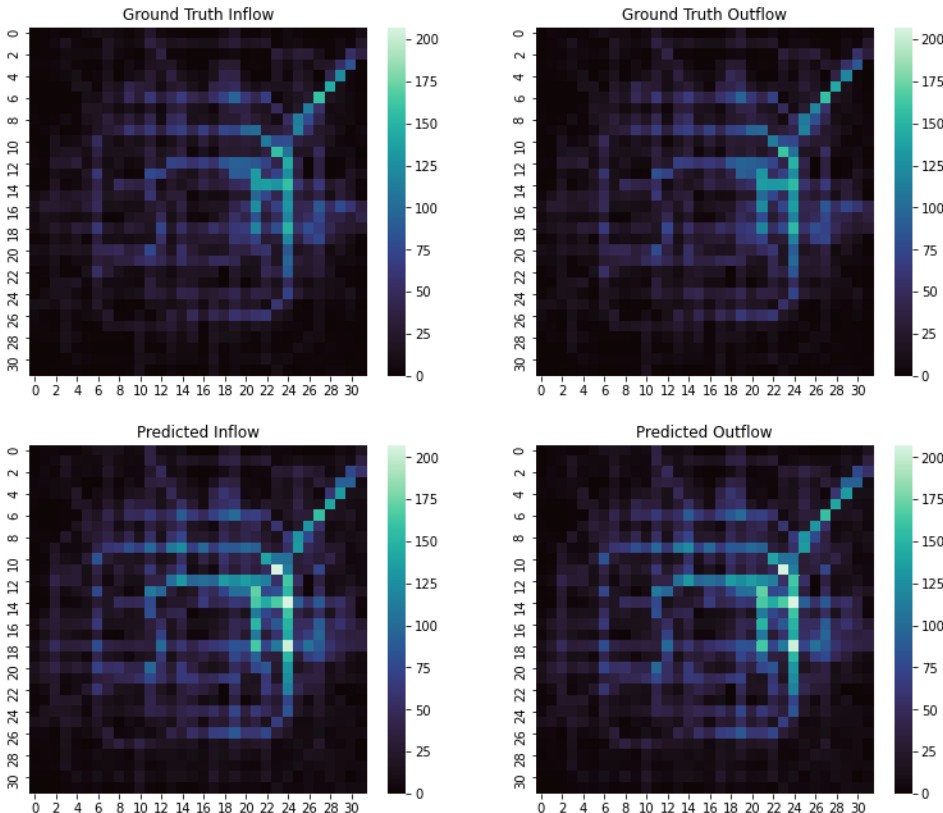

Figure 7: Visualization of predicted results and ground truth traffic flow on TaxiBJ dataset. Figures in the top row are ground truth flows and the bottom ones are the predicted values. STE-TG can achieve accurate traffic flow prediction only using the traffic information of the previous time step and the spatiotemporal representations $e_{stetg}$ of the center grid and the adjacent grids.

We use the top-10 strategy to get the temporal neighboring relations of grids with low distances. Together with the spatial context information $G_{spa}$, $G_{st}$ is built to help the embedding model preserve the spatiotemporal conjunctions between the traffic grids. *The objective function used to optimize the spatiotemporal model remains the same as in the main text.*

**Predictor Design**    The flow of one grid is significantly related to the nature of the grids around it because an outflow from one region can lead to an inflow to neighboring regions. In this work, the spatiotemporal natures of grids and correlations between grids are encoded in the embedding vector $e_{stetg}$. Thus, Given a query $X_{t-1}^{m,n}$, spatiotemporal embeddings of four geographical closest grids and grid $(m, n)$ itself will also be fed into predictor $f_\theta$ to produce the estimation

$$\hat{X}_t^{m,n} = (\hat{x}_t^{in,m,n}, \hat{x}_t^{out,m,n}) = f_\theta(X_{t-1}^{m,n}, \mathbf{e}^{m,n}) \qquad (17)$$

where $\mathbf{e}^{m,n} = (e_{stetg}^{m,n}, e_{stetg}^{neigh})$ and $e_{stetg}^{neigh}$ indicates embeddings of four spatial neighbors of grid $(m, n)$. We use a simple three-layer MLP $f_\theta$ with ReLU activations for implementation, the dimension of hidden state is 256. The loss function used is mean squared error $||\hat{X} - X||_2^2$.

## A.3    EXPERIMENTS

Following the same pipeline we used in POI recommendation task, we first train the spatiotemporal embedding model STE-TG and then optimize the recommender model on the training set. All models are optimized using Adam optimizer with default $\beta$s, the learning rate is set to $1 \times 10^{-3}$. The embedding dimensions $\{d_{spa}, d_{st}\}$ are set to $\{64, 96\}$.

Table 5: Comparisons with baselines on TaxiBJ15 dataset.

| METHOD | RMSE |
|---|---|
| ARIMA | 25.58 |
| SARIMA | 29.11 |
| VAR | 25.59 |
| CNN | 26.08 |
| DeepST-CPTM | 22.59 |
| STE-TG (Ours) | 21.92 |

Table 6: Comparisons with baselines on TaxiBJ dataset.

| METHOD | RMSE |
|---|---|
| HA | 57.69 |
| ARIMA | 22.78 |
| SARIMA | 26.88 |
| LinUOTD | 21.23 |
| ConvLSTM | 19.54 |
| DeepST-CPTM | 18.18 |
| STE-TG (Ours) | 18.03 |

**Metrics**    We measure the traffic flow forecasting performance by Root Mean Square Error (RMSE) as

$$\text{RMSE} \triangleq \sqrt{\frac{1}{t} \sum_i (x_i - \hat{x}_i)^2} \tag{18}$$

where $\hat{x}$ and $x$ are the predicted value and ground truth, respectively and t is the number of all queries.

**Datasets**    We perform experiments on two datasets:

- TaxiBJ15 (Zhang et al., 2016) consists of calculated inflow/outflow based on taxicab GPS data collected in Beijing city in 2015.
- TaxiBJ (Zhang et al., 2017) is an extended version including data from 2013 to 2016.

For TaxiBJ15, data from the last week is used for testing. For TaxiBJ, we choose the last four weeks as the testing data.

**Baselines**    We choose representative baselines for comparison:

- HA: a predict method by the average value of historical values.
- ARIMA (Auto-Regressive Integrated Moving Average)
- SARIMA: seasonal ARIMA
- VAR (Vector Auto-Regressive)
- ST-ANN: It first extracts spatial (nearby 8 regions' values) and temporal (8 previous time intervals) features, then fed into an ANN.
- DeepST (Zhang et al., 2016): a deep neural network (DNN)-based prediction model for spatiotemporal data, which shows state-of-the-art results on crowd flows prediction, we only consider the most potent variant, *i.e.* DeepST-CPTM.
- ST-ResNet (Zhang et al., 2017) ST-ResNet models citywide traffic flow at different times into 2D images to perform prediction.
- LinUOTD (Tong et al., 2017): a linear regression method with a spatio-temporal regularization.
- ConvLSTM (Xingjian et al., 2015): a LSTM variant with convolution modules.

**Results** We listed the comparison results in Table.5 and Table.6. In STE-TG, only the grid map and historical traffic flow are used to perform flow prediction, while existing works usually take meta-data like weather into account. However, we observe that the hastily constructed traffic flow prediction methods STE-TG achieve competitive performance, which also demonstrates the effectiveness of the proposed STE framework.

