# OpenReview forum: "Successive POI Recommendation via Brain-inspired Spatiotemporal Aware Representation"
_ICLR.cc/2022/Conference — ICLR 2022 Submitted_

### Official Review · Reviewer_F7HP · 2021-10-30

**Correctness:** 3
**Technical Novelty And Significance:** 3
**Empirical Novelty And Significance:** 3
**Recommendation:** 6
**Confidence:** 4

**Main Review:**

This paper proposes a spatiotemporal-aware embedding framework and apply to the successive POI recommendation problem. The proposed method, called STEP, is inspired by the mammalian brain entorhinal-hippocampal system that has the conjunctive representation mechanism. Thus, the main motivation of the STEP is to learn the conjunctive representations on the unique spatiotemporal context graphs, which is different from the previous methods dealing with the spatial and temporal information separately.

The experiments are conducted on two real-world datasets, and the experimental results show that the proposed STEP could significantly outperform other baselines including embedding-recommender methods and one-stage methods, although the STEP does not utilize the user preference information.

Overall, the proposed visiting time pattern encoding and spatiotemporal conjunctive embedding learning are interesting and useful, and the paper is well written. The concerns are list below:

1.	For the neighboring timestamps, it is unclear why the weekend time is not be considered.
2.	The paper claims that the proposed framework could be applied to other applications such as the wildlife preservation and urban traffic scheduling problems. However, it is better to give some more details about how to transfer the proposed method to other tasks. For example, the authors could provide a case study experiment on traffic scheduling problem.


**Summary Of The Paper:**

To address the successive POI recommendation problem, this paper proposes a brain-inspired spatiotemporal-aware embedding method, called STEP, which learns the conjunctive representations over the unique spatiotemporal context graphs. The experimental results demonstrate the superiority against other SOTA methods

**Summary Of The Review:**

This paper is technically sound, and the experiments are sufficient. Thus, my rating for this paper is weak accept.

---

> ### Author Response · Authors · 2021-11-17
> **Initial response to #F7HP**
>
> We thank the reviewer for these insightful comments and suggestions. We have grouped our answers into topics with detailed explanations:
>
> **1.** Methods to define temporal adjacency will directly affect the construction of the spatiotemporal context graph. Therefore, the characters of downstream tasks should be taken into account, combined with prior knowledge. This paper introduces a time adjacency definition method suitable for the POI representation task based on the analysis of POI visitings temporal patterns[1]. Specifically, in addition to the basic time adjacency window, we divided the timestamps into weekday/weekend to obtain more fine-grained temporal adjacency relations suitable for the POI downstream tasks like retrieval and recommendation.
>
>
>
> **2.** As the basis of traffic scheduling, it is very important to accurately predict the in/out flow of a region. We perform a case study on urban traffic flow forecasting with STE ,*i.e.,* STE of Traffic Grid (STE-TG). We listed the **RMSE** comparison results on TaxiBJ15 and TaxiBJ datasets in the following tables.  Please refer to **Appendix. A** in the updated paper for more details.
>
>
>
> | Method\Dataset | TaxiBJ15 |
> | -------------- | -------- |
> | ARIMA          | 25.58    |
> | SARIMA         | 29.11    |
> | VAR            | 25.59    |
> | CNN            | 26.08    |
> | DeepST-CPTM    | 22.59    |
> | STE-TG(Ours)   | 21.92    |
>
> | Method\Dataset | TaxiBJ |
> | -------------- | ------ |
> | HA             | 57.69  |
> | ARIMA          | 22.78  |
> | SARIMA         | 26.88  |
> | LinUOTD        | 21.23  |
> | ConvLSTM       | 19.54  |
> | DeepST-CPTM    | 18.18  |
> | STE-TG(Ours)   | 18.03  |

---

### Official Review · Reviewer_a1wE · 2021-11-02

**Correctness:** 3
**Technical Novelty And Significance:** 2
**Empirical Novelty And Significance:** 2
**Recommendation:** 5
**Confidence:** 3

**Main Review:**

Strengths:
The paper proposed a new model to utilize both spatial and temporal information of POIs to make Successive POI Recommendations.

Weaknesses:
1. The paper claims the proposed solution is based on the entorhinal-hippocampal system to process spatial and temporal information. However, spatial and temporal information has long been applied for POI embedding and recommendations. It seems the major difference between the proposed model has the theoretical support from entorhinal-hippocampal systems while others have not, but I cannot find such proof in the paper.

2. It's not clear to find a clear proof or discussion regarding the difference with other models in the use of spatial and temporal information.

3. The writing of Section 3 needs more clarification. It's unclear if the STEP model in Section 3 is learned per sequence, per user, or all records together. The description in Section 3.1 says "Given one POI and its context in the check-in sequence", my understanding is:
The sequential model (Sec. 3.1) G_seq is learned for each sequence, which refers to "one set of continuous checkins of one user in one day".
The spatial model (Sec. 3.2) G_spa is learned for all POIs regarding their coordinates.
The temporal model (Sec. 3.3) is learned from all visits (from different sequences) to each POI.
The spatiotemporal neighboring considers both spatial model and temporal model. Thus, the recommendation highly prefers POIs in the top K closest neighbors which were also visited frequently by all users within a short time window.

**Summary Of The Paper:**

The paper proposed SpatioTemporal aware Embedding framework for POIs (STEP), which considers two types of POI-specific representations: sequential representation and spatiotemporal conjunctive representation. Specifically, the spatiotemporal conjunctive representation represents POIs from spatial and temporal aspects jointly.

**Summary Of The Review:**

The paper needs to clarify the model definition and how each sub-model is integrated.

---

> ### Author Response · Authors · 2021-11-17
> **Initial response to #a1wE**
>
> We thank the reviewer for these insightful comments and suggestions. We have grouped our answers into topics with detailed explanations:
>
> **1.**  Existing works use either contextual information or observation information (such as time interval, spatial location) to represent POIs. Also, they represent items from temporal and spatial dimensions respectively, which leads to sub-optimal predictions. However, the brain-inspired [1,2] STEP  adopts a conjunctive representing method, which is reflected in two aspects. First, representing the conjunction of metric(observation) information and contextual (observation & context conjunction); second, combining information from spatial and temporal dimensions (representing spatiotemporal conjunctions that were built into $G_{st}$​​​​​​​​​​).
>
>
>
> **2.** The STE framework focuses on item-specific representation. For POIs, the STEP representation is POI-specific, and all learning processes are geared towards optimizing POI representation using multiple data (observations and contextual information expressed by context graphs) from different dimensions. In the whole learning process, users are not differentiated, and the user preferences are not considered.
> The sequential model in STEP describes POI from the check-in sequential perspective and results in the sequential adjacency consideration in the recommendation stage.
> On the other hand, the spatiotemporal model in STEP describes POI's spatiotemporal nature and tends to recommend candidates with similar spatiotemporal characteristics (spatiotemporal conjunctions expressed by the geographical and visiting time pattern similar).
>
>
>
> **Refs**
>
> [1] Eichenbaum, Howard. "On the integration of space, time, and memory." *Neuron* 95.5 (2017): 1007-1018.
>
> [2] Jeffery, Kathryn J. "Integration of the sensory inputs to place cells: what, where, why, and how?." *Hippocampus* 17.9 (2007): 775-785.

---

### Official Review · Reviewer_qsPw · 2021-11-02

**Correctness:** 3
**Technical Novelty And Significance:** 3
**Empirical Novelty And Significance:** 2
**Recommendation:** 5
**Confidence:** 4

**Main Review:**

Strengths
1. The writing is easy to follow.
2. The inspiration from the entorhinal-hippocampal system is interesting.

Weakness
1. In the experiment, how to construct user preference embedding for baselines is not clear.
2. What is the difference of the objective between the sequential model and time pattern model? It seems the sequential model can also capture temporal information.

**Summary Of The Paper:**

This paper proposed a successive POI recommendation method inspired by entorhinal-hippocampal system.

**Summary Of The Review:**

The inspiration is interesting. But some parts need further clarification. I suggest "weak reject".

---

> ### Author Response · Authors · 2021-11-17
> **Initial response to #qsPw**
>
> We thank the reviewer for these insightful comments and suggestions. We have grouped our answers into topics with detailed explanations:
>
>
>
> **1.** Some recommendation methods improve the quality of recommendations by assigning unique identifiers to users to depict their preferences. In our comparison table, these methods are marked as 'use user preference embedding' to emphasize the usage of user portrait since STEP only uses anonymous check-in sequences to get embeddings. Although the information from the user-depiction dimension could help the recommendation, it can also lead to privacy issues.
>
> In different methods, the specific realization of obtaining user preference representation/modeling is different,  and detailed descriptions can be found in corresponding papers. Thus, in order to avoid repetitiveness, we do not give a detailed introduction in this paper.
>
>
>
> **2. Distinction between sequential and temporal information**
>
> In the sequential model, the actual time information is actually replaced by the order in the sequence, this is realized by applying a top-k (k=1/2 w, w is the sequential neighboring window width) policy to filter closest POIs as sequential neighbors.  However, the time encoding part of the spatiotemporal model takes a purely temporal consideration, *i.e.*, encodes  timestamps into temporal observation $\mathrm{t}$​​ , and applies an $\epsilon$​​-based ($\epsilon =\mathtt{h}$​​) policy (See Sec.3.2.2) to filter temporal neighbors.

---

### Official Review · Reviewer_sU9T · 2021-11-03

**Correctness:** 2
**Technical Novelty And Significance:** 2
**Empirical Novelty And Significance:** 2
**Recommendation:** 5
**Confidence:** 5

**Main Review:**

The introduced spatiotemporal model of STE, which consists of a grid-cell spatial encoder and a visiting time encoder, is capable of mining the POI-specific spatiotemporal characteristics. Finally, the authors implement successive POI recommendation systems based on the STEP using simple recurrent neural networks as recommenders. Experimental results on Instagram Check-in and Gowalla datasets demonstrate that STEP achieves the state-of-the-art successive POI recommendation performance. However, the novelty of this paper may be over-claimed. My detailed comments are as follows.

Positive points:
1.	The proposed conjunctive representing approach based on a unique spatiotemporal context graph solves the problem of previous spatiotemporal modeling methods in which spatial and temporal information are isolated and represented separately.
2.	The proposed method does not need access to private information such as user preferences.
3.	The experimental results on Instagram Check-in and Gowalla datasets demonstrate the proposed method outperforms baselines.

Negative points:
1.	The motivation and novelty of the proposed method are not convincing.
1)	In the first contribution, the authors claim that they propose this method motivated by the graph-embedding strategy of structural knowledge in the entorhinal-hippocampal system, which is very confusing. The proposed sequential context graph $G_{seq}$ and spatial context graph $G_{spa}$ are very similar to temporal graph $G_t$ and spatial graph $G_s$ in STP-UDGAT[1], respectively. However, the authors do not explain anything about this.
2)	The authors design the grid-cell spatial encoder motivated by grid-cells in the entorhinal-hippocampal system. However, it is worth noting that the formulations of the grid-cell spatial encoder are the same as Space2Vec [2]. The authors only slightly claim that the number of grid scales following the previous work [2].
3)	According to [3], hippocampal place cells encode a geometric representation of space. It is very confusing to claim that hippocampal place cells represent perception conjunctions effectively in the abstract. As shown in Figure 1, why do the authors propose the spatiotemporal conjunctive representation motivated by place cells? More explanations on it may be better.

2.	The authors claim that POI $p_i$ and $p_j$ are spatiotemporal neighboring if they are spatial and temporal neighboring in Sec.3.2.2. However, why do average values of $E_{st}$ per POI exceed those of $E_{spa}$ per POI in Table 3?

Minor issues:

1.	In the Introduction, "Most importantly, we elaborate … an visiting time encoder …" should be "Most importantly, we elaborate … a visiting time encoder …".

2.	In the One-stage methods of Sec.4, “an LSTM variant” and “an LSTM-based method” should be “a LSTM variant” and “a LSTM-based method”, respectively.

References:

[1]	STP-UDGAT : Spatial-Temporal-Preference User Dimensional Graph Attention Network for Next POI Recommendation. In Proceedings of the ACM International Conference on Information & Knowledge Management, pp. 845–854, 2020.
[2]	Multi-Scale Representation Learning for Spatial Feature Distributions using Grid Cells. In Proceedings of the International Conference on Learning Representations, 2020.
[3]	The hippocampus as a predictive map. Nature Neuroscience, 20(11):1643–1653, 2018.


**Summary Of The Paper:**

This paper proposes a SpatioTemporal aware Embedding framework (STE) and applies to POIs (STEP) for successive POI recommendation. Specifically, the proposed STE framework consists of three parts: the context graph building strategies to construct simplified affinity graphs, the spatiotemporal model to extract the spatiotemporal features of check-ins, and the sequential model to extract sequential feature of check-ins.

**Summary Of The Review:**

The introduced spatiotemporal model of STE, which consists of a grid-cell spatial encoder and a visiting time encoder, is capable of mining the POI-specific spatiotemporal characteristics. Finally, the authors implement successive POI recommendation systems based on the STEP using simple recurrent neural networks as recommenders. Experimental results on Instagram Check-in and Gowalla datasets demonstrate that STEP achieves the state-of-the-art successive POI recommendation performance. However, the novelty of this paper seems to be over-claimed.

---

> ### Author Response · Authors · 2021-11-17
> **Initial response to #sU9T**
>
> We thank the reviewer for these insightful comments and suggestions. We have grouped our answers into topics with detailed explanations:
>
>
>
> **1. Context graph construction**
>
> Building affinities between elements into a graph is a very classic idea [1]. Usually, the affinity/similarity is calculated by a distance function. For example, euclidean distance is a common choice for measuring spatial similarity. Also, defining tokens within a specified-width window in sentences/sequences as neighbors are usual in the sentence/sequence-related embedding.
>
> The core of the STEP is a conjunctive embedding model receiving observation&contextual information from spatiotemporal dimension. As the context graphs are used to preserve the qualitative adjacency relations, we assign non-weighted uniform edges to vertices. However, STP-UDGAT constructs spatial/temporal graphs for quantitative calculation in GAT layers to get STA vector  $\vec{o}^A_{t_i}$​​​​​​​. For this purpose, the edge weight of neighboring POI pairs in temporal graph $G_t$​​​​ is assigned as $\frac{1}{\Delta \hat{t}}$​​​​ , *i.e.*, reciprocal averaged time interval and weight of $E_s$​​​  is set to $\frac{1}{\Delta d(v_i, v_j)}$​​​​​​​​​​​​​​​​​​​​.   Therefore, the graph construction in these two works follows some classical operations, but there are apparent differences in the resulting graph and graphs' usage.  In addition, the core of the context graph construction of our work is the construction of spatiotemporal context graph, based on which the STEP model can encode spatiotemporal conjunctions into the vector representation.
>
> We agree with the reviewer that the content of this part should be improved. We have supplemented the discussion about this in the revised version of the paper to improve the content.
>
>
>
> **2. Grid-cell encoder in the spatial sub-model**
>
> We claim to design the spatiotemporal model that consists of a spatial sub-model based on the grid-cell encoder, but not the grid-cell encoder itself, an essential part of the spatiotemporal model in STEP. The grid-cell encoder is based on the spatial representing theorem in [2], which also provides a classic multi-scale spatial representing formulation adopted in [3].
> We have improved the annotation and citation of this part of the content in the revised version.
>
>
>
> **3. Place cells**
>
> While performing remarkable ability to remember 'landmarks' in controlled laboratory environments,  place cells are thought to represent integrations (conjunctions) of sensory inputs from diverse dimensions and sensory & contextual information [4, 5].
>
> We agree with the reviewer that this part could be improved and added relevant explanations in the revised version.
>
>
>
> **4. Fixes**
>
> We have made corrections according to the comments of reviewers, including the issue of stats in Table.3-averaged $|E_{st}|$​​​  caused by not dividing the number of timestamps and other minor expression issues.
>
>
>
> **Refs**
>
> [1] Hsu, Winston H., Lyndon S. Kennedy, and Shih-Fu Chang. "Video search reranking through random walk over document-level context graph." *ACM MM*, 2007.
>
> [2] Ruiqi Gao, Jianwen Xie, and Ying, Nian Zhu, Songchunand Wu. Learning Grid Cells as Vector Representation of Self-Position Coupled with Matrix Representation of Self-Motion. *ICLR*, 2019.
>
> [3] Gengchen Mai, Janowicz Krzysztof, Yan Bo, Zhu Rui, Cai Ling, and Ni Lao. Multi-Scale Representation Learning for Spatial Feature Distributions using Grid Cells. *ICLR*, 2020.
>
> [4] Eichenbaum, Howard. "On the integration of space, time, and memory." *Neuron* 95.5 (2017): 1007-1018.
>
> [5] Jeffery, Kathryn J. "Integration of the sensory inputs to place cells: what, where, why, and how?." *Hippocampus* 17.9 (2007): 775-785.

---

> ### Comment · Reviewer_sU9T · 2021-11-20
> **Further Discussion**
>
> Thanks for the author's responses. The revised version provided an explanation of context graph construction between STEP and STP-UDGAT [1] and solved partial issues about my questions or concerns.
>
> The two main concerns are still remaining:
> 1. The inspiration from the entorhinal-hippocampal system is over-packaged.
> The responses agree that building affinity context graphs between elements are a classic idea [2], so the motivations from the entorhinal-hippocampal system are not convincing. The responses do not explain what is the relationships between the proposed graph-embedding strategy and the entorhinal-hippocampal system. Do any relevant studies clarify the graph-embedding strategy of the entorhinal-hippocampal system?
>
> 2. The claims of place cells are vague.
> The responses claimed that place cells are thought to represent integrations (conjunctions) of sensory inputs from diverse dimensions and sensory & contextual information, which is hard to find support from [3, 4]. But it is too unclear and vague. According to [3], the place cells receive information from many different sensory sources to correctly localize their firing to restricted regions of an environment. As claimed in the conclusion of [3], much integration occurs upstream of the place cells. It is hard to say that place cells are thought to represent integrations (conjunctions). According to [4], the hippocampus map locations in spatially organized environments and map moments in temporally organized experiences. There is no evidence to demonstrate that place cells represent integrations (conjunctions) of space and time information in [4]. According to the authors, they propose the spatiotemporal conjunctive representation motivated by place cells, but this may not be credible.
>
> References:
> [1] STP-UDGAT: Spatial-Temporal-Preference User Dimensional Graph Attention Network for Next POI Recommendation. In Proceedings of the ACM International Conference on Information & Knowledge Management, pp. 845–854, 2020.
> [2] "Video search reranking through random walk over document-level context graph." ACM MM, 2007.
> [3] "Integration of the sensory inputs to place cells: what, where, why, and how?." Hippocampus 17.9 (2007): 775-785.
> [4] "On the integration of space, time, and memory." Neuron 95.5 (2017): 1007-1018.

---

> > ### Author Response · Authors · 2021-11-21
> > **Further response to #sU9T**
> >
> > Thanks for the feedback!
> >
> > The core of this paper is to bridge the spatiotemporal representation mechanism in the entorhinal-hippocampal circuit with practical spatiotemporal representation problems. Albeit the entorhinal-hippocampal structure has long been thought to be highly relevant to the outstanding performance of biological agents on spatiotemporal tasks, the gap between representational hypothesis and specific spatiotemporal tasks remains unbridged. The ability of the entorhinal-hippocampal circuit to represent conjunctions is considered to account for its powerful representation capability, although the clear biological experimental evidence of the spatiotemporal representation theory in the entorhinal-hippocampal circuit is not as thorough as the place cell's memory function for specific locations, as the reviewer said. Therefore, the STE is stated as being inspired by the representation mechanisms rather than mimicking them. In addition, some previous works [1, 2] have proposed model structures for reasoning tasks based on the conjunction representation mechanisms, like the observation-abstract location conjunction representing mechanism [3], in the entorhinal-hippocampal circuit.
> >
> >
> >
> > **Refs**
> >
> > [1] Whittington, J. C. R., et al. "Generalisation of structural knowledge in the hippocampal-entorhinal system." Neural Information Processing Systems (NeurIPS), 2018.
> >
> > [2] Whittington, James CR, et al. "The Tolman-Eichenbaum machine: Unifying space and relational memory through generalization in the hippocampal formation." *Cell* 183.5 (2020): 1249-1263.
> >
> > [3] Komorowski, Robert W., Joseph R. Manns, and Howard Eichenbaum. "Robust conjunctive item–place coding by hippocampal neurons parallels learning what happens where." *Journal of Neuroscience* 29.31 (2009): 9918-9929.

---

### Author Response · Authors · 2021-11-17
**Paper updated !**

Dear Reviewers:

Thanks for the beneficial reviews. We have tried our best to address all the points you have raised. Please find our detailed replies under your reviews.

After revision and supplementation, we have updated our paper, including a *case study* about **traffic flow forecasting** with the proposed STE (presented in **Appendix. A**).

---

### Decision · Program_Chairs · 2022-01-20

**Decision:**

Reject

**Comment:**

Despite some positive points, the criticisms (and overall scores) put this paper below the bar. The reviewers raise issues of novelty, as well as problems with the experiments and argue that some claims are unsupported.